# A New Class of Generalized Fractal and Fractal-Fractional Derivatives with Non-Singular Kernels

Khalid Hattaf [1,2]

1 Equipe de Recherche en Modélisation et Enseignement des Mathématiques (ERMEM), Centre Régional des Métiers de l'Education et de la Formation (CRMEF), Derb Ghalef, Casablanca 20340, Morocco; k.hattaf@yahoo.fr
2 Laboratory of Analysis, Modeling and Simulation (LAMS), Faculty of Sciences Ben M'Sick, Hassan II University of Casablanca, Sidi Othman, Casablanca 20700, Morocco

**Abstract:** The present paper introduces a new class of generalized differential and integral operators. This class includes and generalizes a large number of definitions of fractal-fractional derivatives and integral operators used to model the complex dynamics of many natural and physical phenomena found in diverse fields of science and engineering. Some properties of the newly introduced class are rigorously established. As applications of this new class, two illustrative examples are presented, one for a simple problem and the other for a nonlinear problem modeling the dynamical behavior of a chaotic system.

**Keywords:** fractal-fractional derivative; fractal-fractional integral; numerical scheme; chaotic system

## 1. Introduction

The fractal derivative is a new concept of differentiation that extends the standard derivative for discontinuous fractal media. In the literature, there are various definitions of this new concept. In 2006, Chen [1] introduced the concept of the Hausdorff derivative of a function with respect to a fractal measure $t^\eta$, where $\eta$ is the order of the fractal derivative. This Hausdorff fractal derivative was used to derive a linear anomalous transport-diffusion equation underlying an anomalous diffusion process. He [2] proposed a new fractal derivative for engineering applications in a discontinuous media.

Fractional calculus deals with the generalization of the concepts of differentiation and integration of non-integer orders. This generalization is not merely a purely mathematical curiosity, but it has demonstrated its application in various disciplines such as physics, biology, engineering, and economics. Moreover, fractional differential operators provide an excellent tool for modeling the dynamics of systems possessing memory or hereditary properties. Generally, there are two types of non-local fractional differential operators, ones with singular kernels such as the Caputo derivative [3] and the others with non-singular kernels such as the generalized Hattaf fractional (GHF) derivative [4]. The last fractional operator covers numerous fractional derivatives available in the literature, such as the Caputo–Fabrizio (CF) fractional derivative [5], the Atangana–Baleanu (AB) fractional derivative [6], and the weighted AB fractional derivative [7]. Recently, it was used to model the dynamics of the COVID-19 epidemic using vaccination data in Saudi Arabia based on reported cases [8]. To find the approximate solution for the mathematical models based on the use of fractional derivatives, many numerical methods have been proposed [9–14].

The fractal-fractional derivative is a mathematical concept that combines two different ideas: fractals and fractional derivatives. Fractals are complex geometric patterns that repeat at different scales, while fractional derivatives are a generalization of ordinary derivatives that allow for non-integer orders. The combination of fractal theory and fractional calculus gave rise to new concepts of differentiation and integration. Therefore, several definitions of fractal-fractional derivatives and integrals have been proposed and

developed to solve many real-world problems. Atangana [15] defined six types of fractal-fractional derivatives with exponential decay and Mittag–Leffler kernels by using the Hausdorff fractal derivative. Further, he constructed three fractal-fractional integrals associated with these differential operators with non-singular kernels. Such types of fractal-fractional derivatives and integrals recovered the CF and AB fractional derivatives and integrals. An additional four definitions for fractal-fractional differential and integral operators have been recently introduced in [16] to model the spread of COVID-19.

In more recent years, great importance has been given to fractal-fractional derivatives and their corresponding integrals due to their various applications in modeling several real-life phenomena in many fields, such as epidemiology [17–19], finance [20], ecology [21], and chemistry [22].

The objective of this study is to introduce a new class of fractal-fractional derivatives and integrals based on a new generalized fractal derivative. The importance of this new class is that it encompasses and generalizes the eight definitions for fractal-fractional derivatives with non-singular kernels and the five definitions for fractal-fractional integrals cited above. Furthermore, the newly introduced class includes the GHF derivative that generalizes the CF fractional derivative, the AB fractional derivative, and the weighted AB fractional derivative. In addition, a newly numerical scheme is developed to extend the numerical method presented in [9] for fractal-fractional differential equations (FFDEs), and it is applied to approximate the solution of a model with FFDEs describing the dynamical behavior of a Lorenz chaotic system in order to capture and predict this behavior for different values of fractal and fractional orders.

The outline of this paper is summarized as follows. Section 2 defines the new generalized fractal-fractional derivative in both the Caputo and Riemann–Liouville senses and gives the special cases of such derivatives existing in the literature. Section 3 introduces the generalized fractal-fractional integral associated with this new differential operator and its particular cases. Section 4 demonstrates the applications of our theoretical results through examples with numerical simulations. Finally, Section 5 presents a brief conclusion and some prospects for future research.

## 2. The New Generalized Fractal-Fractional Derivative

The first of this section focuses on the definition of a new generalized fractal derivative. Based on such definition, we develop a new concept of fractal-fractional derivatives in the sense of Caputo and Riemann–Liouville.

**Definition 1.** *The fractal derivative of a function $u(t)$ with respect to a fractal measure $g(\eta, t)$ is defined by*

$$\frac{d_g u(t)}{dt^\eta} = \lim_{\tau \to t} \frac{u(t) - u(\tau)}{g(\eta, t) - g(\eta, \tau)}, \quad \eta > 0. \tag{1}$$

*When $\frac{d_g u(t)}{dt^\eta}$ exists for all $t \in \mathcal{I}$, we say that $u$ is fractal differentiable on the interval $\mathcal{I}$ with order $\eta$.*

It is important to note that when $g(\eta, t) = t^\eta$, Definition 1 reduced to the Hausdorff fractal derivative introduced by Chen [1] in order to model a set of power law scaling phenomena, such as anomalous diffusion, fractional quantum mechanics, and turbulence. If $g(\eta, t) = h(t)$ with $h'(t) > 0$ and $u(t)$ is differentiable, then we obtain the general derivative proposed by Yang in 2019 [23] and Definition 1 becomes

$$\frac{d_g u(t)}{dt^\eta} = \frac{1}{h'(t)} \frac{du(t)}{dt}. \tag{2}$$

Next, we define the new generalized fractal-fractional derivative with non-singular kernel in the sense of Caputo.

**Definition 2.** *Let $p \in [0,1)$, $q, r, \eta > 0$ and $u(t)$ be differentiable in the interval $(a,b)$ and fractal differentiable on $(a,b)$ with order $0 < \eta \leq 1$. Then the generalized fractal-fractional derivative of $u(t)$ of order $p$ in the sense of Caputo with respect to the weight function $w(t)$ is given as follows:*

$$^{FFC}D_{a,t,w}^{p,q,r,\eta}u(t) = \frac{H(p)}{1-p}\frac{1}{w(t)}\int_a^t E_q[-\mu_p(t-\tau)^r]\frac{d_g}{d\tau^\eta}(wu)(\tau)d\tau, \tag{3}$$

*where $w \in C^1(a,b)$, $w, w' > 0$ on $[a,b]$, $H(p)$ is a normalization function such that $H(0) = H(1) = 1$, $\mu_p = \frac{p}{1-p}$ and $E_q(t) = \sum\limits_{k=0}^{+\infty} \frac{t^k}{\Gamma(qk+1)}$ is the Mittag–Leffler function of parameter $q$.*

Definition 2 extends and generalizes various concepts of differentiation existing in the literature. For instance,

1. When $g(\eta, t) = t^\eta$, $w(t) = 1$ and $q = r = 1$, we obtain the fractal-fractional derivative with exponential decay kernel [15] given by

$$^{FFC}D_{a,t,1}^{p,1,1,\eta}f(t) = \frac{H(p)}{1-p}\int_a^t \exp[-\mu_p(t-\tau)]\frac{d_g u(\tau)}{d\tau^\eta}d\tau,$$

   where $\frac{d_g u(\tau)}{d\tau^\eta} = \lim\limits_{t\to\tau}\frac{u(t)-u(\tau)}{t^\eta - \tau^\eta}$.

2. When $g(\eta, t) = t^\eta$, $w(t) = 1$, $H(p) = 1 - p + \frac{p}{\Gamma(p)}$ and $q = r = p$, we obtain the fractal-fractional derivative with generalized Mittag–Leffler kernel [15] given by

$$^{FFC}D_{a,t,1}^{p,p,p,\eta}f(t) = \frac{H(p)}{1-p}\int_a^t E_p[-\mu_p(t-\tau)^p]\frac{d_g u(\tau)}{d\tau^\eta}d\tau.$$

3. When $g(\eta, t) = t^\eta$, $w(t) = 1$, $q = 1$ and $r = 2$, we also obtain the fractal-fractional derivative with exponential decay kernel [15] given by

$$^{FFC}D_{a,t,1}^{p,1,2,\eta}u(t) = \frac{H(p)}{1-p}\int_a^t \exp[-\mu_p(t-\tau)^2]\frac{d_g u(\tau)}{d\tau^\eta}d\tau.$$

4. When $g(\eta, t) = t$, we obtain the generalized Hattaf fractional (GHF) derivative [4] given by

$$^{C}D_{a,t,w}^{p,q,r}u(t) = \frac{H(p)}{1-p}\frac{1}{w(t)}\int_a^t E_q[-\mu_p(t-\tau)^r]\frac{d}{d\tau}(wu)(\tau)d\tau,$$

   which includes the Caputo–Fabrizio fractional derivative [5], the Atangana–Baleanu fractional derivative [6] and the weighted Atangana–Baleanu fractional derivative [7].

   Now, we define the new generalized fractal-fractional derivative using the Riemann–Liouville sense.

**Definition 3.** *Let $p \in [0,1)$, $q, r, \eta > 0$ and $u(t)$ be continous in the interval $(a,b)$ and fractal differentiable on $(a,b)$ with order $0 < \eta \leq 1$. Then the generalized fractal-fractional derivative of $u(t)$ of order $p$ in the sense of Riemann–Liouville with respect to the weight function $w(t)$ is given as follows:*

$$^{FFR}D_{a,t,w}^{p,q,r,\eta}u(t) = \frac{H(p)}{1-p}\frac{1}{w(t)}\frac{d_g}{dt^\eta}\int_a^t E_q[-\mu_p(t-\tau)^r]w(\tau)u(\tau)d\tau. \tag{4}$$

Obviously, Definition 3 also includes the three recent fractal-fractional differential operators introduced by Atangana [16] to model the spread of COVID-19, it suffices to choose $g(\eta, t) = \frac{t^{2-\eta}}{2-\eta}$. In this case, the generalized fractal derivative given in Definition 1 becomes

$$\frac{d_g u(t)}{dt^\eta} = \lim_{\tau \to t} \frac{u(t) - u(\tau)}{t^{2-\eta} - \tau^{2-\eta}} (2 - \eta). \tag{5}$$

On the other hand, the new GHF derivative in the Riemann–Liouville sense [4] is recovered if $\eta = 1$ and $g(\eta, t) = t$.

**Theorem 1.** *Let $\frac{\partial g(\eta,t)}{\partial t}$ be exist and not zero. Then*

$$^{FFR}D_{a,t,w}^{p,q,r,\eta}u(t) = \left(\frac{\partial g(\eta, t)}{\partial t}\right)^{-1} \left[{}^C D_{a,t,w}^{p,q,r}u(t) + \frac{H(p)}{1-p}\frac{1}{w(t)}E_q[-\mu_p(t-a)^r](wu)(a)\right]. \tag{6}$$

**Proof.** We have

$$
\begin{aligned}
^{FFR}D_{a,t,w}^{p,q,r,\eta}u(t) &= \frac{H(p)}{1-p}\frac{1}{w(t)}\frac{d_g}{dt^\eta}\int_a^t E_q[-\mu_p(t-\tau)^r]w(\tau)u(\tau)d\tau \\
&= \frac{H(p)}{1-p}\frac{1}{w(t)}\left(\frac{\partial g(\eta, t)}{\partial t}\right)^{-1}\frac{d}{dt}\int_a^t E_q[-\mu_p(t-\tau)^r]w(\tau)u(\tau)d\tau.
\end{aligned}
$$

By applying Theorem 1 of [4], we deduce that

$$^{FFR}D_{a,t,w}^{p,q,r,\eta}u(t) = \left(\frac{\partial g(\eta, t)}{\partial t}\right)^{-1}\left[{}^C D_{a,t,w}^{p,q,r}u(t) + \frac{H(p)}{1-p}\frac{1}{w(t)}E_q[-\mu_p(t-a)^r](wu)(a)\right].$$

This ends the proof. $\square$

## 3. The Generalized Fractal-Fractional Integral

In this section, we first solve the following fractal-fractional differential equation:

$$^{FFR}D_{0,t,w}^{p,q,q,\eta}v(t) = u(t), \tag{7}$$

which leads to

$$\left(\frac{\partial g(\eta, t)}{\partial t}\right)^{-1}{}^R D_{0,t,w}^{p,q,q}v(t) = u(t).$$

Then , we find

$$^R D_{0,t,w}^{p,q,q}v(t) = u(t)\frac{\partial g(\eta, t)}{\partial t}. \tag{8}$$

According to Theorem 3 of [4], we have

$$v(t) = \frac{1-p}{H(p)}u(t)\frac{\partial g(\eta, t)}{\partial t} + \frac{p}{H(p)\Gamma(q)w(t)}\int_0^t (t-\tau)^{q-1}w(\tau)u(\tau)\frac{\partial g(\eta, \tau)}{\partial \tau}d\tau. \tag{9}$$

**Definition 4.** *If $r = q$, then we define the generalized fractal-fractional integral associated with the new fractal-fractional derivative as follows*

$$^{FF}I_{0,t,w}^{p,q,q,\eta}u(t) = \frac{1-p}{H(p)}u(t)\frac{\partial g(\eta, t)}{\partial t} + \frac{p}{H(p)\Gamma(q)w(t)}\int_0^t (t-\tau)^{q-1}w(\tau)u(\tau)\frac{\partial g(\eta, \tau)}{\partial \tau}d\tau. \tag{10}$$

**Remark 1.** *The associate integral defined above includes a variety of fractal-fractional integral operators. For example,*

**(i)** *If $g(\eta, t) = t^\eta$, then (10) becomes*

$$^{FF}I_{0,t,w}^{p,q,q,\eta}u(t) = \frac{\eta(1-p)}{H(p)}t^{\eta-1}u(t) + \frac{p\eta}{H(p)\Gamma(q)w(t)}\int_0^t (t-\tau)^{q-1}\tau^{\eta-1}w(\tau)u(\tau)d\tau. \tag{11}$$

*Further, the two fractal-fractional integrals for exponential decay kernel and Mittag–Leffler kernel [15] are recovered, it suffices to take in (11) $w(t) = 1$ with $q = 1$ for the fist integral, and $q = p$ for the second one.*

**(ii)** *If $g(\eta, t) = \frac{t^{2-\eta}}{2-\eta}$, then (10) becomes*

$$^{FF}I_{0,t,w}^{p,q,q,\eta}u(t) = \frac{1-p}{H(p)}t^{1-\eta}u(t) + \frac{p}{H(p)\Gamma(q)w(t)}\int_0^t (t-\tau)^{q-1}\tau^{1-\eta}w(\tau)u(\tau)d\tau. \quad (12)$$

*Hence, Equation (12) includes the recent fractal-fractional integral for exponential decay kernel [16] when $w(t) = 1$ and $q = 1$, as well as the fractal-fractional integral for Mittag–Leffler kernel [16] when $w(t) = 1$ and $q = p$.*

**(iii)** *If $g(\eta, t) = t$, then (10) reduced to the new GHF integral presented in [4].*

## 4. Examples

This section provides two examples to give an idea of the applicability of our results. For simplicity, let us denote $^{FFR}D_{0,t,w}^{p,q,\tilde{q},\eta}$ by $^{\mathcal{FFR}}\mathcal{D}_{t,w}^{p,q,\eta}$.

**Example 1.** *Consider the following fractal-fractional differential equation*

$$^{\mathcal{FFR}}\mathcal{D}_{t,w}^{p,q,\eta}u(t) = t. \quad (13)$$

By Definition 3, we obtain

$$^{R}D_{0,t,w}^{p,q,q}u(t) = t\frac{\partial g(\eta, t)}{\partial t}. \quad (14)$$

According to Theorem 3 of [4], we have

$$u(t) = \frac{(1-p)t}{H(p)}\frac{\partial g(\eta, t)}{\partial t} + \frac{p}{H(p)\Gamma(q)w(t)}\int_0^t (t-\tau)^{q-1}w(\tau)\tau\frac{\partial g(\eta, \tau)}{\partial \tau}d\tau. \quad (15)$$

For $w(t) = 1$ and $g(\eta, t) = t^\eta$, the above equation becomes as follows

$$u(t) = \frac{\eta(1-p)t^\eta}{H(p)} + \frac{p\eta}{H(p)\Gamma(q)}\int_0^t (t-\tau)^{q-1}\tau^\eta d\tau.$$

Since $\frac{1}{\Gamma(q)}\int_0^t (t-\tau)^{q-1}\tau^\eta d\tau = \frac{\Gamma(\eta+1)t^{q+\eta}}{\Gamma(q+\eta+1)}$, we have

$$u(t) = \frac{\eta\Gamma(\eta+1)}{H(p)}\left[\frac{1-p}{\Gamma(\eta+1)} + \frac{pt^q}{\Gamma(q+\eta+1)}\right]t^\eta. \quad (16)$$

**Remark 2.**

**(i)** *For $q = 1$, Equation (13) becomes a problem with CF derivative. In this case, the solution of (13) becomes*

$$u(t) = \frac{\eta\Gamma(\eta+1)}{H(p)}\left[\frac{1-p}{\Gamma(\eta+1)} + \frac{pt}{\Gamma(\eta+2)}\right]t^\eta. \quad (17)$$

**(ii)** *For $q = p$, Equation (13) becomes a problem with AB derivative. In this case, the solution of (13) becomes*

$$u(t) = \frac{\eta\Gamma(\eta+1)}{H(p)}\left[\frac{1-p}{\Gamma(\eta+1)} + \frac{pt^p}{\Gamma(p+\eta+1)}\right]t^\eta. \quad (18)$$

The two problems with CF and AB derivatives presented in Remark 2 have been solved recently in [24] by means of the Laplace transform with several steps and computations. However, our approach is simple, and it requires only two steps, the transformation of

the problem to be solved into a problem with the GHF derivative and the application of Theorem 3 in [4].

Figures 1–4 illustrate the solutions of (13) in the case of GHF, AB, and CF derivatives for different values of $p$, $q$, and $\eta$. We show that the three solutions coincide when fractal and fractional orders tend to one.

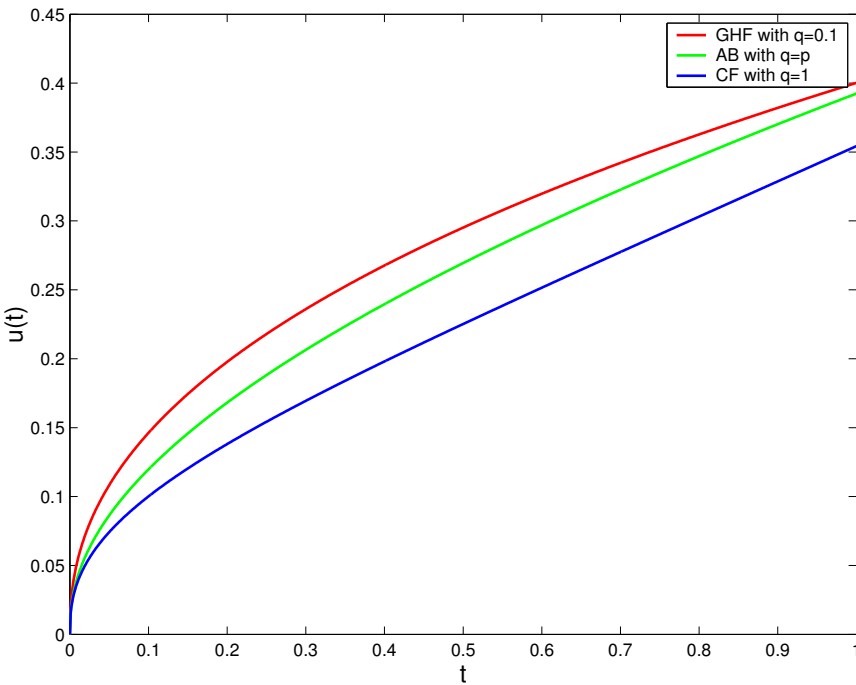

**Figure 1.** Solutions of (13) for GHF, AB, and CF derivatives for different values of $q$ and $p = \eta = 0.4$.

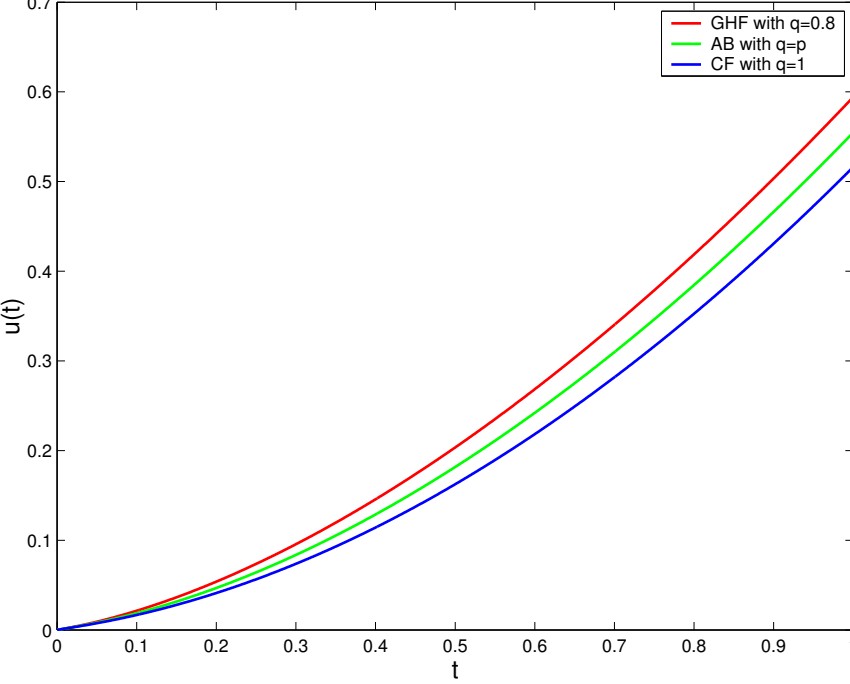

**Figure 2.** Solutions of (13) for GHF, AB, and CF derivatives for different values of $q$ and $p = \eta = 0.9$.

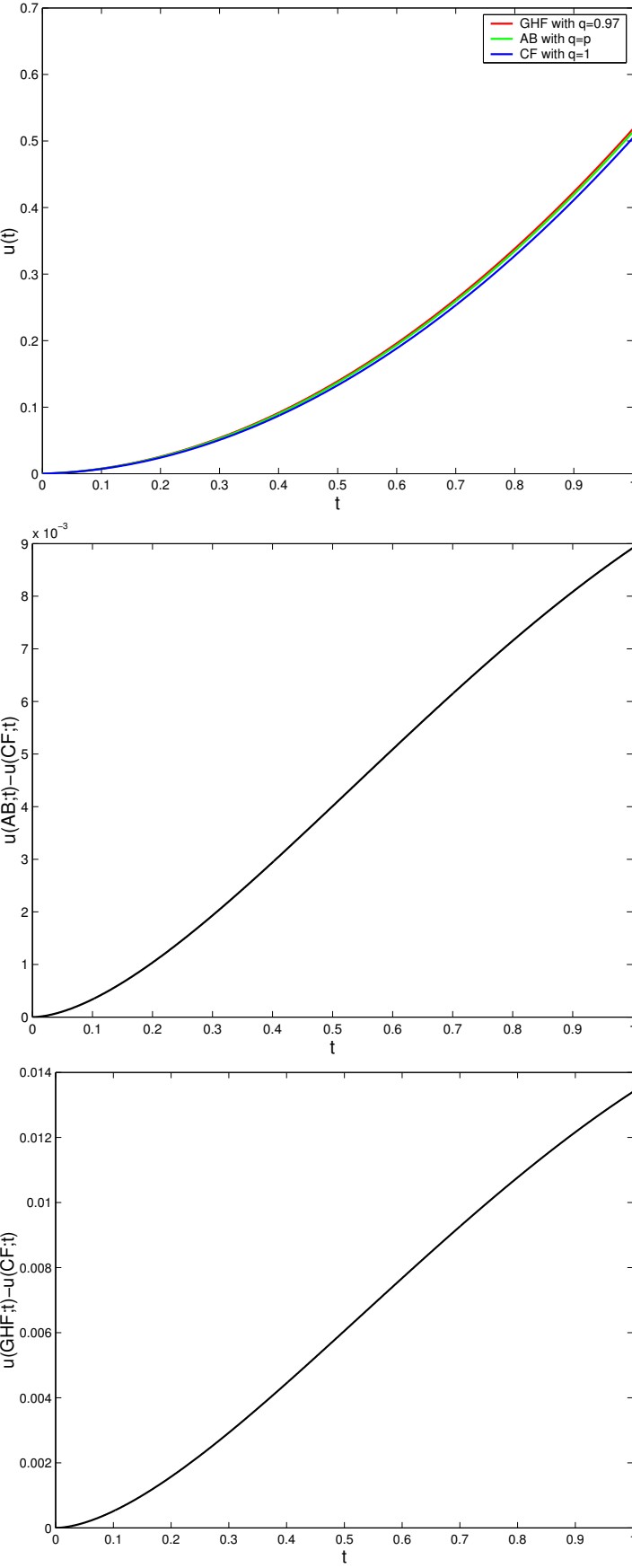

**Figure 3.** Solutions of (13) for GHF, AB, and CF derivatives for different values of $q$ and $p = \eta = 0.98$.

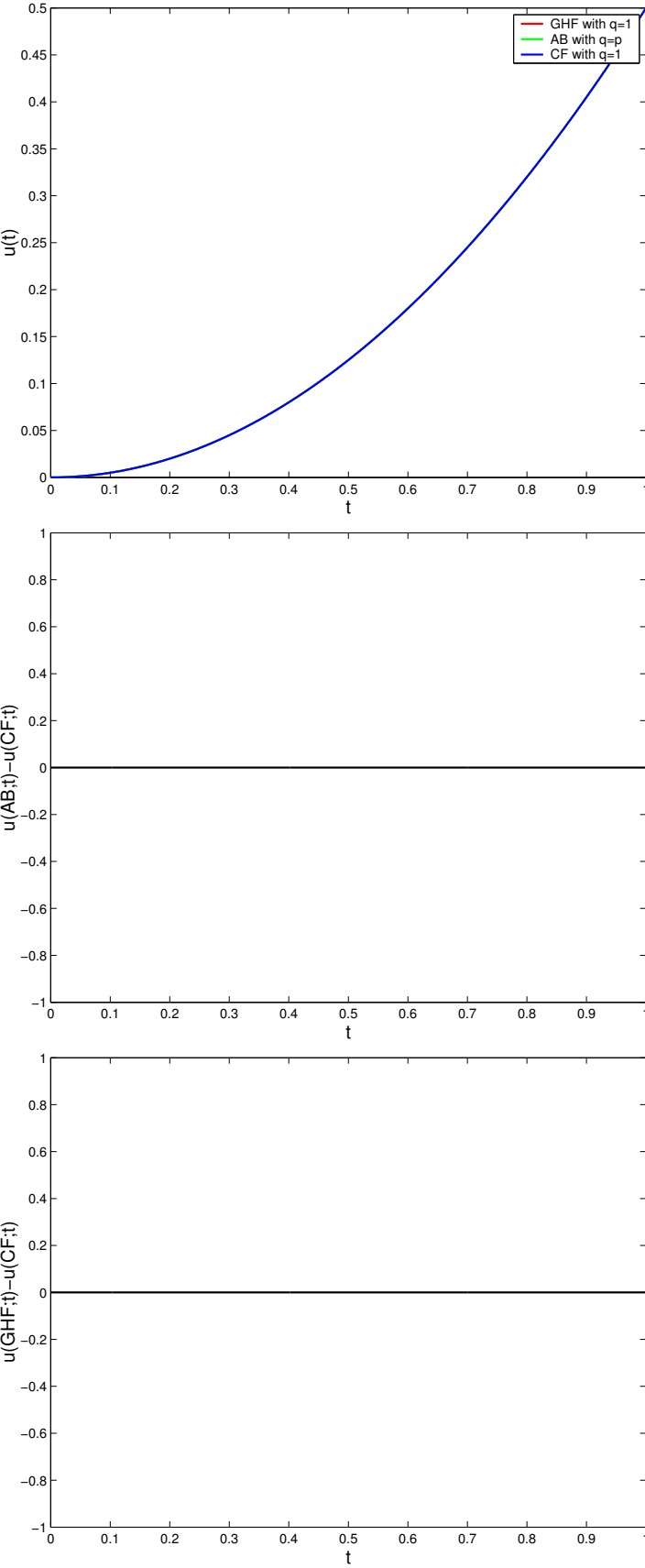

**Figure 4.** Solutions of (13) for GHF, AB, and CF derivatives for different values of *q* and $p = \eta = 1$.

**Example 2.** *Consider the following nonlinear Lorenz chaotic system described by fractal-fractional differential equations as follows:*

$$
\begin{cases}
\mathcal{FFR}\mathcal{D}_{t,w}^{p,q,\eta}x(t) &= f_1(t,x(t),y(t),z(t)), \\
\mathcal{FFR}\mathcal{D}_{t,w}^{p,q,\eta}y(t) &= f_2(t,x(t),y(t),z(t)), \\
\mathcal{FFR}\mathcal{D}_{t,w}^{p,q,\eta}z(t) &= f_3(t,x(t),y(t),z(t)),
\end{cases}
\tag{19}
$$

*where $x(t)$, $y(t)$ and $z(t)$ are the state variables of the system, and*

$$
\begin{aligned}
f_1(t,x(t),y(t),z(t)) &= \sigma(y(t)-x(t)), \\
f_2(t,x(t),y(t),z(t)) &= x(t)(\rho-z(t))-y(t), \\
f_3(t,x(t),y(t),z(t)) &= x(t)y(t)-\delta z(t),
\end{aligned}
\tag{20}
$$

*with $\sigma$, $\rho$ and $\delta$ are the parameters of system.*

Similarly to above, system (19) can be rewritten as follows

$$
\begin{cases}
{}^{R}D_{0,t,w}^{p,q,q}x(t) &= \frac{\partial g(\eta,t)}{\partial t}f_1(t,x(t),y(t),z(t)), \\
{}^{R}D_{0,t,w}^{p,q,q}y(t) &= \frac{\partial g(\eta,t)}{\partial t}f_2(t,x(t),y(t),z(t)), \\
{}^{R}D_{0,t,w}^{p,q,q}z(t) &= \frac{\partial g(\eta,t)}{\partial t}f_3(t,x(t),y(t),z(t)).
\end{cases}
\tag{21}
$$

Substituting ${}^{R}D_{0,t,w}^{p,q,q}$ by ${}^{C}D_{0,t,w}^{p,q,q}$ in order to make the use of the integer-order initial conditions and operating the Hattaf fractional integral on both sides of (21), we have

$$
\begin{aligned}
x(t) &= \frac{x(0)w(0)}{w(t)} + \frac{1-p}{H(p)}F_1(t,x(t),y(t),z(t)) \\
&\quad + \frac{p}{N(p)\Gamma(q)w(t)}\int_0^t (t-\tau)^{q-1}w(\tau)F_1(\tau,x(\tau),y(\tau),z(\tau))d\tau, \\
y(t) &= \frac{y(0)w(0)}{w(t)} + \frac{1-p}{H(p)}F_2(t,x(t),y(t),z(t)) \\
&\quad + \frac{p}{N(p)\Gamma(q)w(t)}\int_0^t (t-\tau)^{q-1}w(\tau)F_2(\tau,x(\tau),y(\tau),z(\tau))d\tau, \\
z(t) &= \frac{z(0)w(0)}{w(t)} + \frac{1-p}{H(p)}F_3(t,x(t),y(t),z(t)) \\
&\quad + \frac{p}{N(p)\Gamma(q)w(t)}\int_0^t (t-\tau)^{q-1}w(\tau)F_3(\tau,x(\tau),y(\tau),z(\tau))d\tau,
\end{aligned}
$$

where $F_i(t,x(t),y(t),z(t)) = \frac{\partial g(\eta,t)}{\partial t}f_i(t,x(t),y(t),z(t))$ for $i=1,2,3$.

Let $t_n = nh$, where $n \in \mathbf{N}$ and $h$ is the time step duration. Based on the numerical scheme [9], we find

$$
\begin{aligned}
x(t_{n+1}) &= \frac{x(0)w(0)}{w(t_n)} + \frac{1-p}{H(p)} F_1\big(t_n, x(t_n), y(t_n), z(t_n)\big) \\
&\quad + \frac{ph^q}{H(p)\Gamma(q+1)w(t_n)} \sum_{k=0}^{n} w(t_k) F_1\big(t_k, x(t_k), y(t_k), z(t_k)\big) \mathcal{A}_{n,k}^q, \\
y(t_{n+1}) &= \frac{y(0)w(0)}{w(t_n)} + \frac{1-p}{H(p)} F_2\big(t_n, x(t_n), y(t_n), z(t_n)\big) \\
&\quad + \frac{ph^q}{H(p)\Gamma(q+1)w(t_n)} \sum_{k=0}^{n} w(t_k) F_2\big(t_k, x(t_k), y(t_k), z(t_k)\big) \mathcal{A}_{n,k}^q, \\
z(t_{n+1}) &= \frac{z(0)w(0)}{w(t_n)} + \frac{1-p}{H(p)} F_3\big(t_n, x(t_n), y(t_n), z(t_n)\big) \\
&\quad + \frac{ph^q}{H(p)\Gamma(q+1)w(t_n)} \sum_{k=0}^{n} w(t_k) F_3\big(t_k, x(t_k), y(t_k), z(t_k)\big) \mathcal{A}_{n,k}^q,
\end{aligned} \tag{22}
$$

where

$$
\mathcal{A}_{n,k}^q = (n-k+1)^q - (n-k)^q. \tag{23}
$$

For numerical simulation, we choose $\sigma = 10$, $\rho = 28$, $\delta = \frac{8}{3}$, $g(\eta, t) = \frac{t^{2-\eta}}{2-\eta}$, $w(t) = 1$ and $H(p) = 1 - p + \frac{p}{\Gamma(p)}$. Therefore, Figures 5–9 demonstrate the dynamics of a Lorenz chaotic system (19) for different values of fractal and fractional orders.

To analyze numerically the influence of parameter $p$ on Lorenz chaotic system, we use the bifurcation diagram (see, Figure 10) for $q = \eta = 1$ and $p \in [0.98, 1]$ with the incremental value of $p$ is 0.0002. For example, the solution converges to positive equilibrium when $p \leq 0.991$ (see, Figures 11 and 12), and to negative equilibrium when $p = 0.992$ (see, Figure 13). However, it becomes unstable when $p = 0.993$ (see, Figure 14). Similarly, we can investigate the impact of the parameters $q$ and $\eta$ on the dynamics of the Lorenz chaotic system by means of the same method.

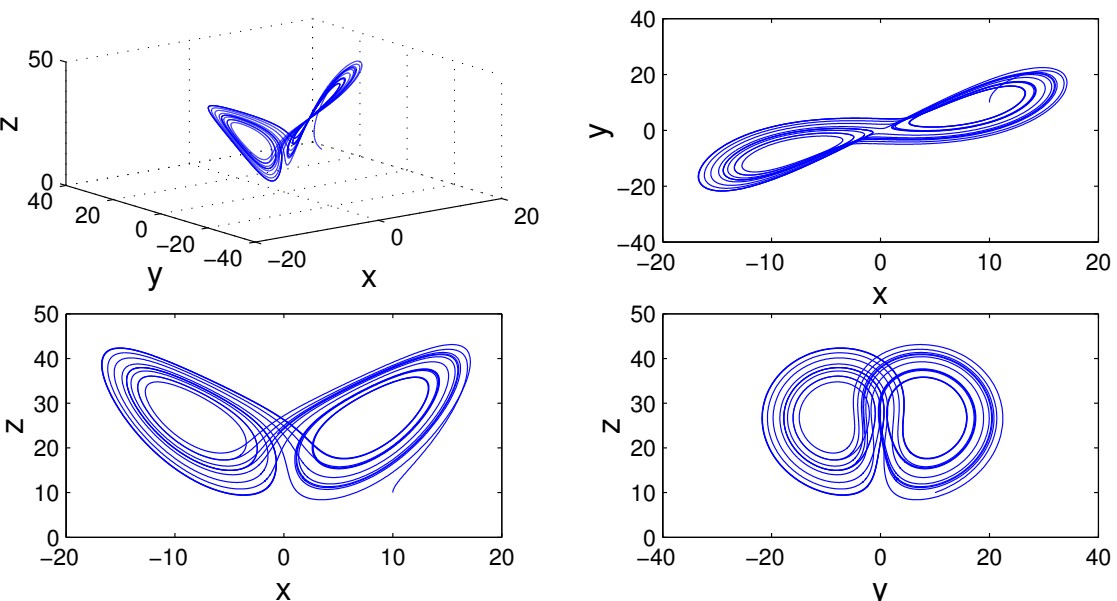

**Figure 5.** Dynamical behavior of the Lorenz chaotic system (19) for $p = 1$, $q = 1$ and $\eta = 1$.

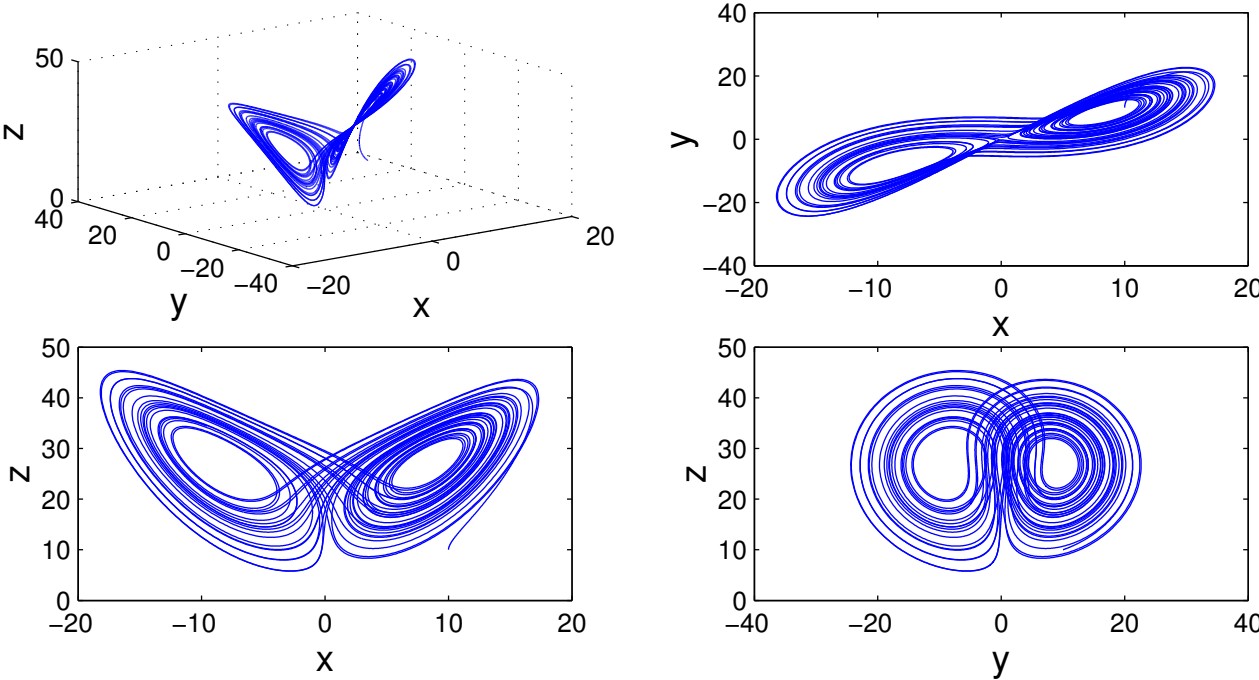

**Figure 6.** Dynamical behavior of the Lorenz chaotic system (19) for $p = 1$, $q = 1$ and $\eta = 0.7$.

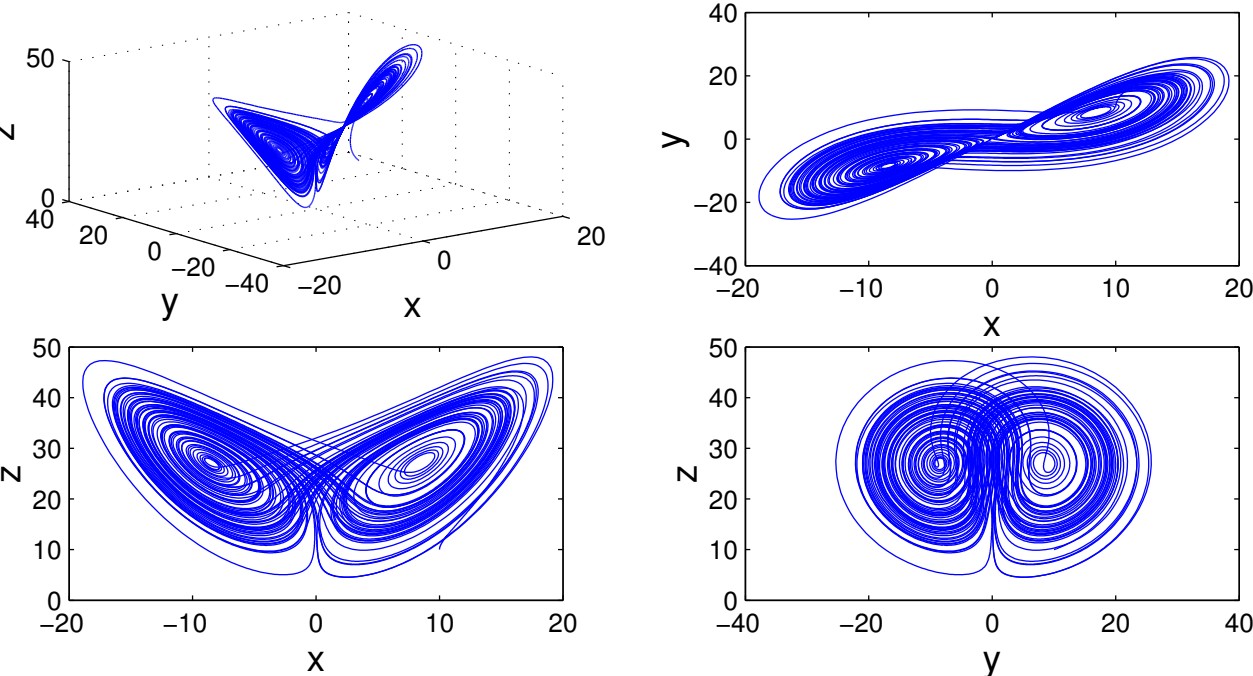

**Figure 7.** Dynamical behavior of the Lorenz chaotic system (19) for $p = 1$, $q = 1$ and $\eta = 0.4$.

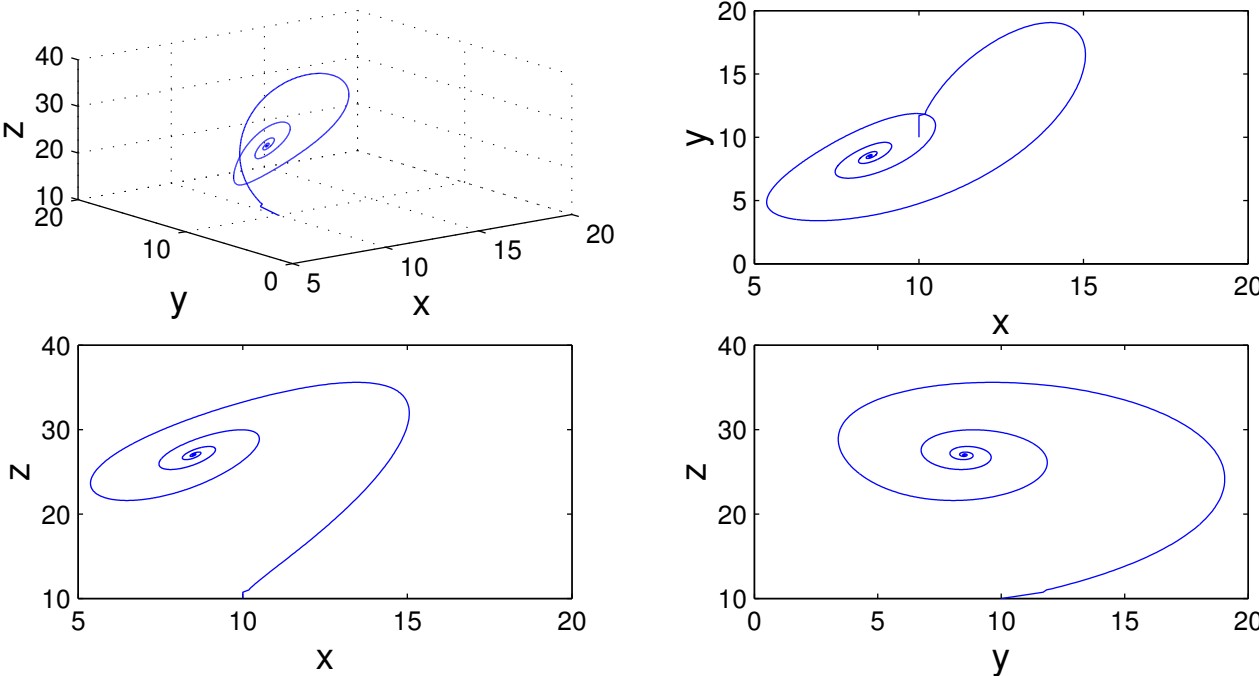

**Figure 8.** Dynamical behavior of the Lorenz chaotic system (19) for $p = 0.99$, $q = 0.95$ and $\eta = 1$.

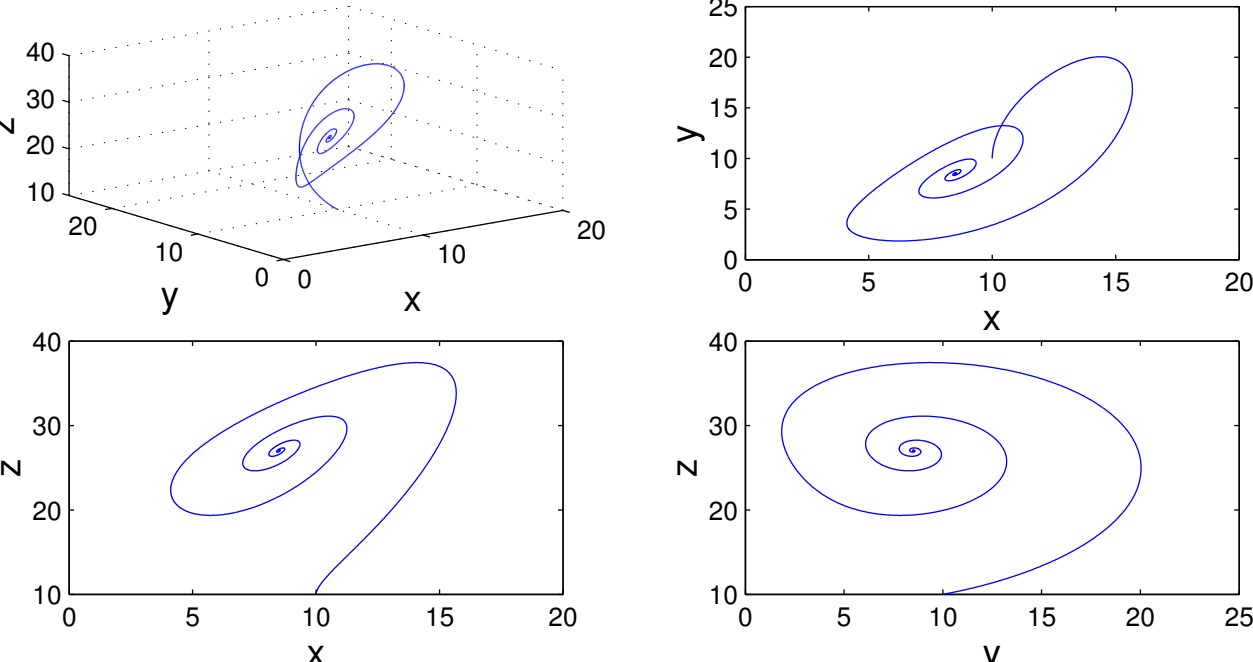

**Figure 9.** Dynamical behavior of the Lorenz chaotic system (19) for $p = 0.99$, $q = 0.95$ and $\eta = 0.5$.

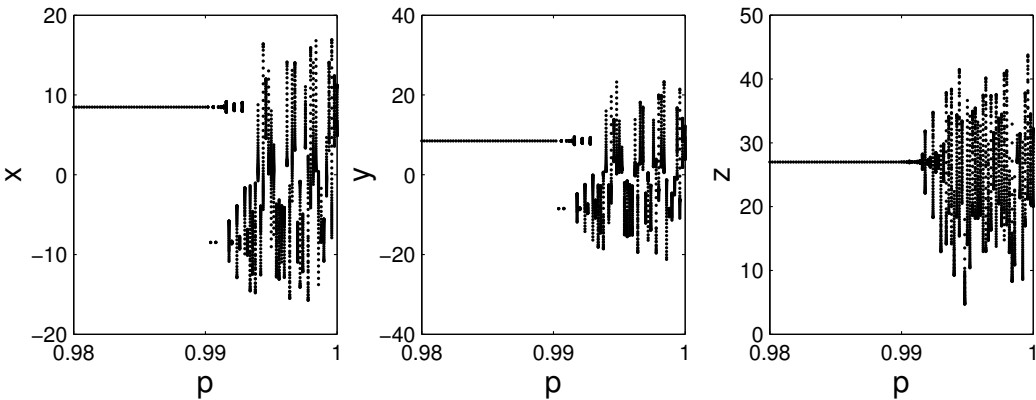

**Figure 10.** Bifurcation diagram of the Lorenz chaotic system (19) with $p \in [0.98, 1]$.

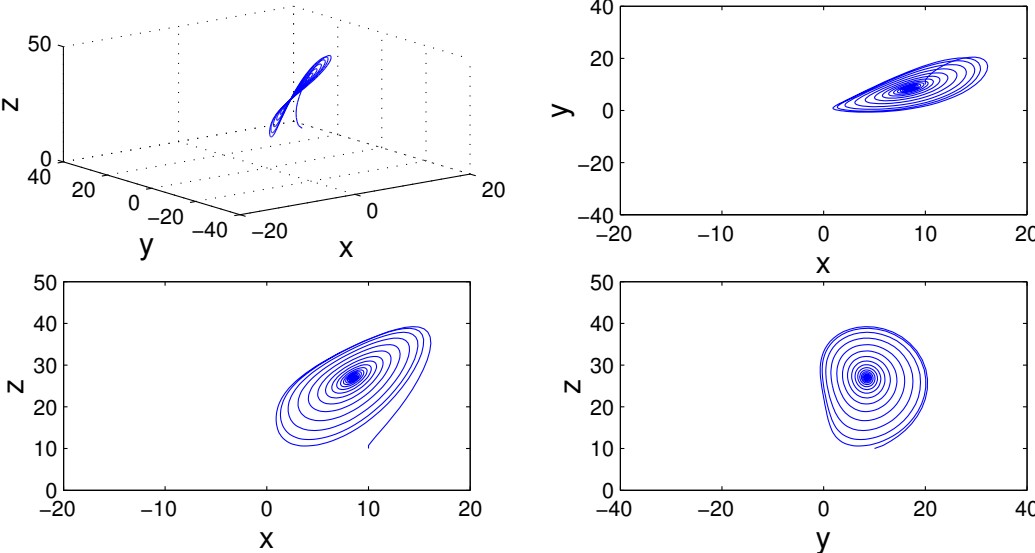

**Figure 11.** Dynamical behavior of the Lorenz chaotic system (19) for $p = 0.990$, $q = 1$ and $\eta = 1$.

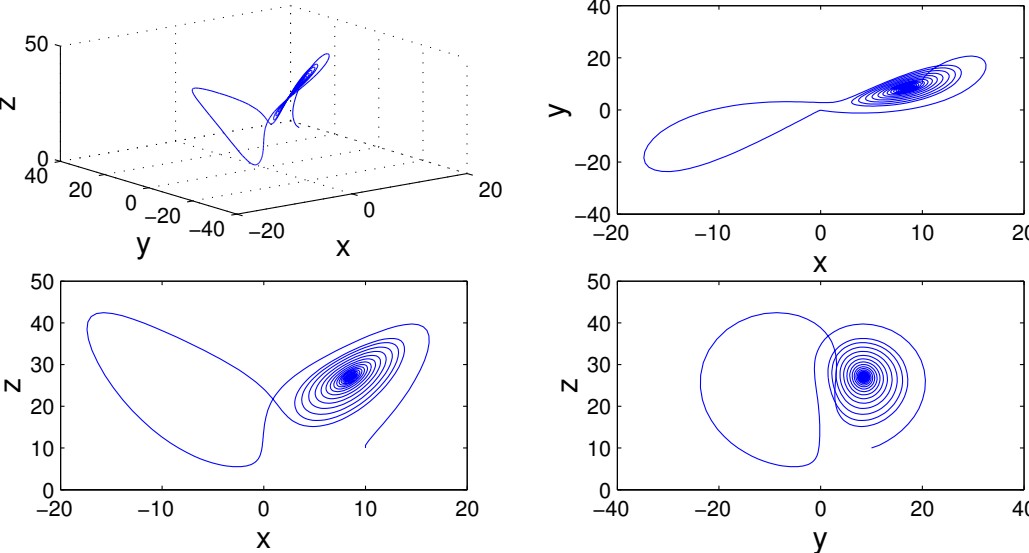

**Figure 12.** Dynamical behavior of the Lorenz chaotic system (19) for $p = 0.991$, $q = 1$ and $\eta = 1$.

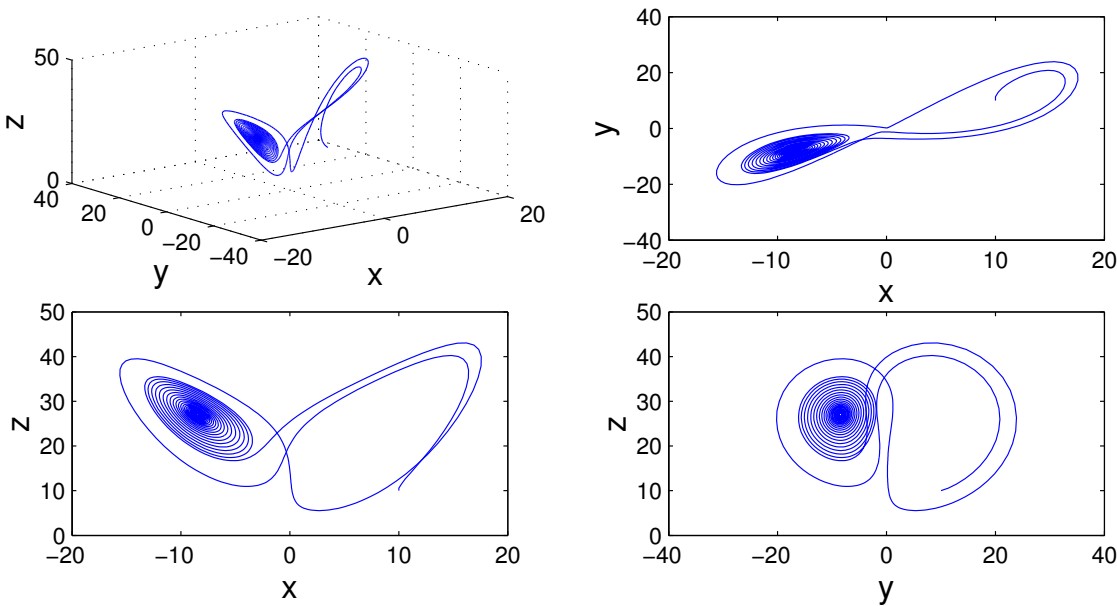

**Figure 13.** Dynamical behavior of the Lorenz chaotic system (19) for $p = 0.992$, $q = 1$ and $\eta = 1$.

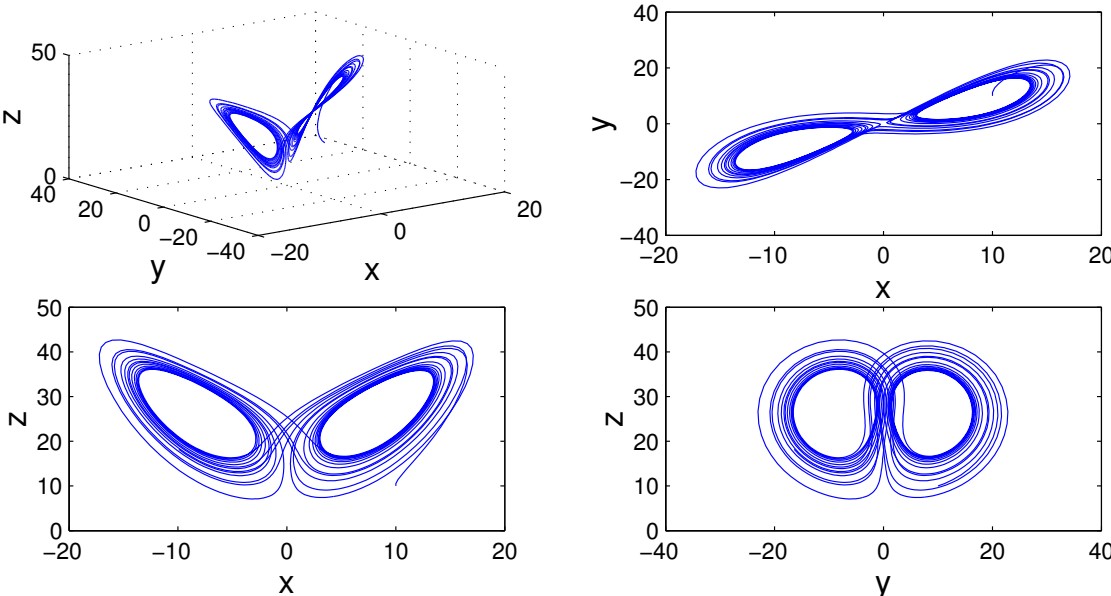

**Figure 14.** Dynamical behavior of the Lorenz chaotic system (19) for $p = 0.993$, $q = 1$ and $\eta = 1$.

## 5. Conclusions

In this paper, we have introduced a new class of fractal-fractional derivatives and integrals by means of a new generalized fractal derivative that includes the Hausdorff fractal derivative used to describe the anomalous diffusion process. The newly introduced class of differential and integral operators extends and generalizes the eight definitions for fractal-fractional derivatives with non-singular kernels and the five definitions for fractal-fractional integrals recently presented in [15,16]. In addition, this class recovers the new GHF derivative that encompasses the most existing fractional derivatives such as the CF fractional derivative [5], the AB fractional derivative [6] and the weighted AB fractional derivative [7]. Moreover, we have developed a new numerical method for solving fractal-fractional differential equations involving the new generalized fractal-fractional derivative.

By comparing the new generalized fractal-fractional derivative developed in this study with other existing fractal-fractional operators, we conclude that such a new fractal-

fractional derivative is a non-local operator having a general fractal derivative and a non-singular kernel, in which the recent forms defined in [15,16] become special cases. However, it is more interesting to solve problems and establish pure and applied results by means of a general fractal-fractional operator.

The stability analysis and the development of other numerical schemes for fractal-fractional differential equations with the new generalized fractal-fractional derivative, as well as the modeling of real-world phenomena having memory and fractal properties will be the prospects for future research.

**Funding:** This research received no external funding.

**Institutional Review Board Statement:** Not applicable.

**Informed Consent Statement:** Not applicable.

**Data Availability Statement:** Not applicable.

**Acknowledgments:** The author would like to thank the editor and anonymous referees for their very helpful comments and suggestions that greatly improved the quality of this study.

**Conflicts of Interest:** The author declares no conflict of interest.

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
