# Peer review of "A New Class of Generalized Fractal and Fractal-Fractional Derivatives with Non-Singular Kernels"

_fractalfract, doi:10.3390/fractalfract7050395_

Round 1
Reviewer 1 Report
In this paper, the influence of related parameters (p, q) on Loren chaotic system is analyzed only by using phase diagram. It is suggested to add methods, such as adding bifurcation diagram, Lyapunov exponent or complexity, to further show the influence of parameters on the system.
In this paper, the influence of related parameters (p, q) on Loren chaotic system is analyzed only by using phase diagram. It is suggested to add methods, such as adding bifurcation diagram, Lyapunov exponent or complexity, to further show the influence of parameters on the system.
Reviewer 2 Report
The article is not written in MDPI format. The paper needs to be formatted to match the journal format.
Typos in line 14: "diff¨Ä±¿½rentiation."
I recommend Figures 6 and 7 to be a gradient color to be visually easier to understand.
What is your study's applicability to real-world problems?
English language and grammar mistakes were found in the paper and will need revision.
Reviewer 3 Report
1. What is the definition of Fractal-fractional derivative? What is its difference with non-fractal fractional derivative?
2. Is the linear example 1 fractal-fractional derivative?
3. What will happen if the system is also singular? Check some fractional singular systems.
4. Where is the conclusion of the study? What are the differences of this study with previous works? Highlight challenges, contributions and results of the work.
A moderate editing of English language is needed.
Reviewer 4 Report
The paper deals with an important task. It has a scientific novelty. The description of practical significance of the results needs to be improved. It has a logical structure. The paper is technically sound. The proposed approach is logical. The results presentation section needs to be improved.
Suggestions:
1. In formulating the aims of the research, it is advisable to clearly indicate what exactly will allow to taken into account the use of fractional derivatives in the model. And confirm this while analyzing the results obtained.
2. The authors proposed a numerical scheme for finding an approximate solution. However, the results of the study of the stability of the solution depending on n are not shown.
3. Validation of the obtained results was not carried out, at least in a partial case.
4. Figures 1 - 8 show the results of the numerical implementation of the proposed scheme, but there is no analysis and description of these results.
5. The obtained results are given for different values of the fractal fractional derivative indicator. In particular, α = 0.4, α = 0.9 and so on. By what principle were such values chosen, and what is their meaning in the context of the model?
6. In the figures, it is advisable to indicate the units of measurement along each of the coordinate axes.
7. It would be appropriate to expand the overview section with a review of methods for finding the solution for the mathematical models based on the use of fractional order derivatives, such as approximation using splines, using the finite element method and others.
8. Conclusion section should be extended using:
· numerical results obtained in the paper;
· limitations of the proposed approach;
· prospects for future research.

Reviewer 5 Report
Fractional calculus has been used in different scenarios as a powerful tool to investigate various aspects of the system and connect them with experimental data. After the fractional differential operators with singular kernels, extensions to nonsingular kernels have been considered. Other generalizations have also been analyzed, such as the fractal operators. The author considers an extension of these operators to a new class of differential operators. The results are interesting and may be helpful for research in this field. I suggest the author improve the introduction with a complete discussion of the fractional operators. In particular, it would be interesting if the author could explain the motivation for considering a general fractional operator in more detail. The physics or mathematics behind these extensions could be discussed as a benefit to the readers. In this sense, the author presented some examples where a comparison with the previous fractional differential operators was missed and could be performed in detail to clarify the differences and the new aspects gained with this approach. A conclusion could be incorporated into the manuscript to discuss the previous points.
Reviewer 6 Report
(1) About Quality of English Language.
(1.1) I don't feel qualified to judge about the English language and style.
(1.2) in References, Page 14, bibitems 10, 11, 12, 14:
Something is wrong with the code table,
see specifically lines 184, 187, 191, 194.
(1.3) Page 14, Line 165: substitute / Differention / Differentiation /
(1.4) Page 6, Line 112: substitute / sufices / suffices /
(1.5) Page 1, Line 14: something is wrong with the code table (see ' of diff')
(2) About Figures.
(2.0) in Legend of Fig.1-4 substitute /CHF/GHF/
(as is in the captions of figures).
(2.1) Add a sub-figure to Fig. 3 (also for Fig. 4), where the differences
u(AB;t)-u(CF;t) and u(GHF;t)-u(CF;t) are drawn along the ordinate axis Оy.
(2.2) For Figues 5,6,7:
Let axis Y in sub-figure (Top-Left) to be within [-40.40]
Let axis Z in sub-figures (BL,BR)=(Bottom-Left and Right) to be in [0,50].
(2.3) For Figure 8:
Let axis Z in sub-figure (Top-Left) to be within [10,40]
(2.4) For Figure 9:
Let axis Z in sub-figure (Top-Left) to be within [10,40]
Let axis Y in sub-figures (TL,TR)=(Top-Left and Right) to be within [0,25]
(3) About Equations.
(3.1) Only equations 10-13,19,21 are cited (have references to) in the body of
article. Missing references to equations 1-9,14-18,20,22,23.
The reviewer's recommendation is:
Either to cite (to mention) them in the body of the article or remove redundant numbering and correct references in the text where necessary.
(3.2) Some of the equations go beyond the margins,
see Page 5 (6) and Line 2 before Section 3, also
(10) after Line 107, (11), (12) - Page 6, after Lines 110, 114.
The reviewer's recommendation is: to move them to the next row immediately
after the part with the equal sign '='.
(3.3) Increase by at least 2pt '\\[2pt]' in LaTeX, the vertical spacing between
lines in equations (19) and (21)

(1) About Quality of English Language.
(1.1) I don't feel qualified to judge about the English language and style.
(1.2) in References, Page 14, bibitems 10, 11, 12, 14:
Something is wrong with the code table,
see specifically lines 184, 187, 191, 194.
(1.3) Page 14, Line 165: substitute / Differention / Differentiation /
(1.4) Page 6, Line 112: substitute / sufices / suffices /
(1.5) Page 1, Line 14: something is wrong with the code table (see ' of diff')
Round 2
Reviewer 2 Report
- Why isn’t the new generalized fractal-fractional operator compared with other generalized techniques to adhere its importance or superiority? Please, if possible, compare with other generalized operators.
NA
Reviewer 5 Report
I have analyzed the revised version of the manuscript and considered it suitable for publication.
